# Overall Survival and Prognostic Factors among Older Patients with Metastatic Pancreatic Cancer: A Retrospective Analysis Using a Hospital Database

**DOI:** 10.3390/cancers14051105

**Published:** 2022-02-22

**Authors:** Catherine Conti, Frédéric Pamoukdjian, Thomas Aparicio, Soraya Mebarki, Johanne Poisson, Gilles Manceau, Julien Taieb, Bastien Rance, Sandrine Katsahian, Anaïs Charles-Nelson, Elena Paillaud

**Affiliations:** 1Service de Gériatrie, Hôpital Européen Georges Pompidou, Paris Cancer Institute CARPEM, APHP, 75015 Paris, France; catherine.conti@aphp.fr (C.C.); soraya.mebarki@aphp.fr (S.M.); johanne.poisson@aphp.fr (J.P.); 2Service de Médecine Gériatrique, Hôpital Avicenne, APHP, 93000 Bobigny, France; 3Inserm UMR_S942, Cardiovascular Markers in Stressed Conditions, MASCOT, Université Sorbonne Paris Nord, 93000 Bobigny, France; 4Service de Gastroentérologie, Hôpital Saint Louis, APHP, Université de Paris, 75010 Paris, France; thomas.aparicio@aphp.fr; 5Service de Chirurgie Digestive et Oncologique, Hôpital Européen Georges Pompidou, Paris Cancer Institute CARPEM, University of Paris, APHP, 75015 Paris, France; gilles.manceau@aphp.fr; 6Service d’Hépatogastroentérologie, Hôpital Européen Georges Pompidou, Paris Cancer Institute CARPEM, Université de Paris, APHP, 75015 Paris, France; julien.taieb@aphp.fr; 7Département d’Informatique Médicale, Hôpital Européen Georges Pompidou, Paris Cancer Institute CARPEM, Université de Paris, APHP, 75015 Paris, France; bastien.rance@aphp.fr; 8INSERM UMR_S 1138 équipe 22, Centre de Recherche des Cordeliers, 75006 Paris, France; 9Unité de Recherche Clinique, Hôpital Européen Georges Pompidou, APHP, 75015 Paris, France; sandrine.kastahian@aphp.fr (S.K.); anais.charles-nelson@aphp.fr (A.C.-N.); 10INSERM, Centre d’Investigation Clinique 1418 (CIC1418) Épidémiologie Clinique, 75908 Paris, France; 11Clinical, Epidemiology and Ageing, IMRB-UPEC/Inserm U955, Université Paris-Est Creteil, 94000 Creteil, France

**Keywords:** pancreatic cancer, chemotherapy, older adults, overall survival, prognostic factors, diabetes, functional status

## Abstract

**Simple Summary:**

The benefits of standard treatments in metastatic pancreatic cancer (mPC) in terms of overall survival (OS) remain to date unclear, especially after 70 years. Alongside geriatric and oncologic parameters, we showed that the gemcitabine + nab-paclitaxel regimen and anti-diabetic therapy were significantly associated with a better OS, while impaired functional status, the liver metastases and high neutrophil count were associated with a worse OS in older adults with mPC. We confirm the feasibility and efficacy of chemotherapy in older adults with mPC.

**Abstract:**

Pre-therapeutic factors associated with overall survival (OS) among older patients ≥70 years with metastatic pancreatic cancer (mPC) are not known. This was a retrospective single-centre cohort study in Paris including 159 consecutive older patients with mPC between 2000 and 2018. Alongside geriatric parameters, specific comorbidities, cancer-related data and chemotherapy regimens were retrieved. Cox multivariate models were run to assess predictors for OS. The median age was 80 years, 52% were women, 21.5% had diabetes, and 48% had pancreatic head cancer and 72% liver metastases. 62% of the patients (*n* = 99) received chemotherapy, among which the gemcitabine + nab-paclitaxel (GnP) regimen was the most frequent (72%). Median OS [95%CI] was 7.40 [5.60–10.0] and 1.40 [0.90–2.20] months respectively for patients with and without chemotherapy. The GnP regimen (aHR [95%CI] = 0.47 [0.25–0.89], *p* = 0.02) and diabetes (aHR = 0.44 [0.24–0.77], *p* = 0.004) (or anti-diabetic therapy) were multivariate protective factors for death, while ECOG-PS, liver metastases, and the neutrophil cell count were multivariate risk factors for death. In the chemotherapy group, ECOG-PS, number of metastatic sites and the GnP remained significantly associated with OS. Our study confirms the feasibility and efficacy of chemotherapy and the protective effects of diabetes among older patients with mPC.

## 1. Introduction

With a median age of 69 years and 73 years at diagnosis for men and women respectively, pancreatic cancer is a disease of the elderly. Despite advances in cancer treatment, pancreatic cancer, mainly diagnosed at a metastatic stage, remains one of the solid cancers worldwide with the poorest diagnosis, and its incidence and mortality increase with age [1].

The standard treatment for metastatic pancreatic cancer (mPC) has for a long time been gemcitabine alone [2]. In 2011, compared to gemcitabine alone, the PRODIGE-4/ACCORD-11 clinical trial showed the superiority of the FOLFIRINOX regimen (combining 5- fluorouracil + leucovorin + oxaliplatin + irinotecan) as first-line treatment for overall survival in mPC (11.1 months vs. 6.8 months; HR = 0.57, *p* < 0.001) [3]. In addition, compared to gemcitabine alone, the MPACT clinical trial [4] in 2013 showed the superiority of a regimen combining gemcitabine + nab-paclitaxel (GnP) first line for overall survival (8.5 months vs. 6.7 months; HR = 0.72, *p* < 0.001). These two regimens are now considered the standard care for mPC. However, older adults ≥70 years were not represented in these two trials.

While some older patients could benefit from chemotherapy, the benefits of standard treatments in mPC in terms of survival remain to date unclear, especially after 70 years [5,6]. Indeed, regarding elderly patients, the main concern is patient heterogeneity in terms of comorbidities, dependency, malnutrition or functional status, each of these factors being liable to be associated with poor outcomes in the course of cancer treatment [7]. To clarify this point, the Geriatric Assessment (GA), which is a multidimensional health assessment for older adults, was initially recommended to guide cancer-treatment decisions [8], but no study to date has assessed the predictive value of the GA for overall survival among older patients with mPC.

The identification of pre-therapeutic factors taking into account geriatric and oncologic parameters that are associated with shorter survival could help select older patients who could benefit from standard chemotherapy.

Here, we aimed to assess pre-therapeutic factors that were associated with overall survival during first-line treatment among older adults ≥70 years with mPC, and to compare patients who received chemotherapy with those who did not.

## 2. Materials and Methods

In this study, we followed the recommendations of the STrengthening the Reporting of OBservational studies in Epidemiology method (STROBE) for the reporting of observational epidemiological studies [9].

### 2.1. Study Design and Population

A retrospective analysis of older patients with mPC and treated between 1 October 2000 and 31 October 2018 in a single centre in Paris (Georges Pompidou European Hospital) was performed. All consecutive older adults ≥70 years with histologically confirmed primitive pancreatic cancer at metastatic stage at diagnosis and with no previous treatment were included. Neuroendocrine tumors and pancreatic lymphoma were excluded.

All consecutive eligible patients were retrieved from medical records using the International Classification of Diseases, v10, if they were coded as follows: “malignant neoplasm of the pancreas” (C25) + “secondary malignant neoplasm of the liver and intrahepatic bile duct” (C78.7) + “secondary malignant neoplasm in unspecified location “(C79.9) [10].

The inclusion date was the date of the diagnosis.

The study was approved by the local ethics committee (CERAPHP; reference: 10 July 2021).

### 2.2. Data Collection

At the time of the first oncology consultation, demographic data including age, gender, and marital status (married/single) was recorded.

Cancer-related data was collected in the course of the first line of treatment: date of diagnosis, location (tail, body, isthmus, head), number of metastatic sites, Eastern Cooperative Oncology Group Performance Status (ECOG-PS), and presence of ascites at diagnosis or endobiliary stenting. The number of metastatic sites was determined using the CT-scan reports or medical records or during weekly multidisciplinary meeting reports. Cancer treatment was classified as follows: single gemcitabine or gemcitabine + nab-paclitaxel (GnP) or 5-fluorouracil/leucovorin + oxaliplatin (FOLFOX) or fluorouracil/leucovorin + irinotecan + oxaliplatin (FOLFIRINOX) or 5-fluorouracil simplified (5-FU) or capecitabine alone or exclusively supportive care. Among the chemotherapy regimens, single gemcitabine was chosen as the reference.

Geriatric parameters were retrospectively retrieved from medical records and included the following 7 domains: total comorbidities (Charlson’s updated comorbidity index ≥ 1) and specific comorbidities (diabetes, heart failure, renal failure, chronic respiratory failure, liver failure, history of stroke); poly-medication (≥5 drugs a day) [11]; dependency (Activity of Daily Living scale (ADL) ≤ 5/6) [12]; walking limitations (walks alone or with help) [13]; malnutrition (weight loss ≥ 5% in the previous year; body mass index (BMI) < 21 kg/m^2^; albumin level < 35 g/L) [14,15]; and history of depressed mood or cognitive impairment.

Covariates were as follows: neutrophil cell count (G/L), lymphocyte cell count (G/L), neutrophil to lymphocyte ratio (NLR), haemoglobin level (anaemia defined as below 12 g/dL); C-Reactive Protein (CRP) (mg/L), Ca-19.9 (kUI/L), and total bilirubin level (μmol/L). We also considered type of diabetes treatment as covariates as follows: no treatment, insulin alone, oral anti-diabetic drugs or insulin + oral anti-diabetic drugs.

### 2.3. Outcomes

The primary outcome was overall survival (OS). Vital status was determined from the medical records or the public records office.

The secondary outcome was the profile of patients with chemotherapy.

### 2.4. Statistical Analysis

Categorical variables were described as numbers (%) and quantitative variables were described as a means ± SD or median (min-max), as appropriate.

Categorical variables were compared with Pearson’s chi-square test or Fisher’s exact test, as appropriate. Continuous variables were compared using Student’s *t* test or Wilcoxon’s test, as appropriate.

Overall survival: Univariate survival curves were estimated according to the Kaplan-Meier method for chemotherapy status (yes/no) in the whole cohort, then for number of metastasis sites (≤1 vs. >1), for ECOG-PS (0–2 vs. 3–4), and for chemotherapy regimens (by reference for single gemcitabine) in the chemotherapy subgroup. Cox uni- and multivariate proportional hazards regression models were run to assess pre-therapeutic factors associated with overall survival. The model assumptions were verified. Variables yielding *p*-values (Wald test) under 0.20 and with less than 15% of missing data in the univariate analysis were considered for inclusion in the multivariate analysis. A stepwise selection process based on the lowest Akaïke criterion was performed to retain the final multivariate models. Additional stratified analyses according to chemotherapy status (yes/no) were also performed. The association between pre-therapeutic factors and overall survival was expressed using adjusted hazard ratios (aHRs) and a 95% CI.

All tests were two-sided, and the threshold for statistical significance was set at *p* < 0.05. The data was analysed using R statistical software (version4.0.2, R Foundation for Statistical Computing, Vienna, Austria; http://www.r-project.org, accessed on 10 January 2022).

## 3. Results

### 3.1. Patients

During the study period, 159 consecutive patients aged 70 and over with mPC were retrospectively included in this study (Figure 1).

### 3.2. Baseline Characteristics and Comparison between Patients with and without Chemotherapy

Table 1 shows the baseline characteristics of the 159 patients with mPC. The mean age was 80.2 ± 6.3 years. A majority of patients were women (52%), had pancreatic head cancer (48%) mainly with liver metastases (72%) and a low ECOG-PS score (≥3). According to the tools and thresholds used, the level of impairment in the domains explored by the geriatric parameters varied from 9% (cognitive impairment) to 66% (weight loss and albumin <35 g/L). Among comorbidities, heart failure (33.5%) and diabetes (21.5%) were the most frequent. Among anti-diabetic treatments, oral anti-diabetic drugs concerned 19/34 (56%) of the diabetic patients. The distribution of oral anti-diabetic drugs was available for 17 of them as follows: 14/17 (82%) were treated with metformin, 7/17 (41%) with glinides, and 4/17 (23.5%) with sulfonamides.

The majority of patients received a chemotherapy regimen (*n* = 99, 62%), mainly gemcitabine alone or GnP (*n* = 72; 72%). Patients with a chemotherapy regimen were significantly younger and more often married, there was a significantly lower proportion of liver metastases, and lower scores for ECOG-PS, ADL-dependency, and walking limitations, and a significantly smaller proportion of haemoglobin levels <12 g/dL (anaemia), and a significantly lower CAR, NLR and total bilirubin values (Table 1).

### 3.3. Pre-Therapeutic Factors Associated with Overall Survival among Older Patients with Metastatic Pancreatic Cancer

Median OS [95%CI] was 7.40 [5.60–10.0] and 1.40 [0.90–2.20] months respectively among patients with and without chemotherapy (Figure 2). Depending on the chemotherapy regimen, median OS was 4.80 [3.90–7.90], 13.5 [11.6–21.3], 3.65 [2.00-NA], and 10.1 [8.80-NA] for gemcitabine alone, GnP, FOLFOX and FOLFIRINOX, respectively. The median chemotherapy duration was 64 days. Depending on the chemotherapy regimen, the median time of exposure was 38.0, 108.0, 60.5 and 64.5 days for gemcitabine alone, GnP, FOLFOX and FOLFIRINOX, respectively. There were no significant differences between median times of exposure to chemotherapy regimens (*p* = 0.20). For diabetes status, there was no significant difference across the different chemotherapy regimens in terms of median chemotherapy duration (*p* = 0.90).

Whole cohort (Table 2): In univariate analysis, age, ECOG-PS, the ADL scale, walking limitations, diabetes, tumour site, liver metastases, the NLR and the type of chemotherapy regimen were significantly associated with overall survival. Due to a collinearity between ECOG-PS and ADL, we retained ECOG-PS for multivariate analyses. Due to a collinearity between diabetes and anti-diabetic therapy, we provided two multivariate models, one including diabetes (model 1), the other including anti-diabetic therapy as covariate (model 2). In both multivariate models, while chemotherapy regimens and diabetes or anti-diabetic therapy were significant protective factors for mortality, the ECOG-PS, liver metastases, and neutrophil cell count (but not the NLR) were significant risk factors for mortality. There was no significant interaction among the multivariate predictors.

Patients with chemotherapy (Table 3): In univariate analysis, ECOG-PS, ADL scores, walking limitations, diabetes, primary tumour site, metastasis site, the number of metastatic sites, and the GnP regimen (by reference for single gemcitabine) were significantly associated with overall survival. Figure 3 shows univariate survival curves according to the number of metastasis sites, ECOG-PS, and chemotherapy regimens (by reference for single gemcitabine). Due to a collinearity between diabetes and anti-diabetic therapy, we provided two multivariate models, one including diabetes (model 1), the other including anti-diabetic therapy as covariate (model 2). Regardless the final multivariate model, ECOG-PS, the number of metastatic sites and the GnP regimen (by reference for single gemcitabine) remained significantly associated with overall survival.

## 4. Discussion

In this original retrospective study including 159 consecutive older adults with a median age of 80 years and with metastatic pancreatic cancer (mPC) not previously treated, 62% received chemotherapy, mainly gemcitabine alone or the GnP regimen. Alongside the geriatric parameters, we found that dependency (ADL and ECOG-PS), liver metastasis, high neutrophil cell count and diabetes (or anti-diabetic therapy) were independent pre-therapeutic factors associated with overall survival (OS). In a stratified analysis of patients with chemotherapy, ECOG-PS, the number of metastatic sites and the GnP regimen remained significantly associated with OS.

The main finding of our study is that pre-therapeutic diabetes and anti-diabetic therapy were significantly associated with better OS among older adults ≥70 years with mPC in the overall cohort but not in the chemotherapy subgroup (probably because of a lack of power). Although we did not distinguish between type-1 and type-2 diabetes, because of the retrospective study design, this result could be explained by the significantly lower prevalence of sarcopenia among type-2 diabetic patients treated with metformin, which was recently highlighted in a large meta-analysis involving 16,800 patients [16]. Indeed, in our study, 82% of the diabetic patients were treated with metformin which is in line with the literature [17,18]. The age-related loss of skeletal muscle mass and function (sarcopenia) is a well-known independent risk factor for shorter survival among cancer patients [19]. The role of metformin combines both anti-cancer activity and protein synthesis via direct and indirect pathways [20]. Metformin was also found to be associated with a significant benefit for overall survival in a large meta-analysis involving 24,178 cancer patients, leading the authors to recommend the use of metformin as an adjuvant treatment for cancer [21]. More recently, the protective effect of metformin on mortality among pancreatic cancer patients was specifically confirmed in a large meta-analysis involving 38,772 patients with a pooled Hazard Ratio of 0.81 [95%CI: 0.70–0.91]. Nevertheless, in line with our study, this protective effect was not found in the chemotherapy subgroup, with a pooled HR of 0.99 [95%CI: 0.67–1.30] [22]. Other anti-diabetic drugs including insulin could also have a benefit on muscle mass and function, but there have been no studies to date supporting this hypothesis among older adults, particularly on account of the supposed effect of age-related insulin resistance [23].

Another strength of our study was that for the first time we reported the predictive value of geriatric parameters among older patients with mPC. Indeed, as previously reported in other cancers, we were able to confirm that ADL-dependency and walking limitations were geriatric parameters that were significantly associated with shorter survival during the first line of treatment for mPC [7,24]. We were also able to provide a comprehensive characterization of the geriatric health status in this specific population and before cancer-treatment decisions, which was consistent with other large observational studies in geriatric oncology [7,25]. Moreover, in agreement with previous studies, we found that the neutrophil count and the number of metastatic sites were independently associated with OS [26,27].

Regarding the benefit of chemotherapy on overall survival among older patients with mPC, our study results are consistent with other studies on younger patients since we found similar median survival rates associated with chemotherapy regimens [3,4]. Specifically, we found that the use of a chemotherapy regimen in older patients with metastatic pancreatic cancer, added a significant benefit in terms of overall survival, even after adjustment for functional and nutritional status. The best regimen that we found in terms of feasibility and efficacy remains the GnP regimen (median OS = 13.5 months), but the small number of patients probably underestimated the impact of other poly-chemotherapy regimens. Indeed, recent retrospective studies have reported the feasibility and efficacy of the main regimens among older adults with mPC: Pignon et al. reported a median OS of 8.0 months for the GnP regimen with no significant differences between younger and older patients (< or ≥75 years) [28]; Mizrahi et al. reported a median OS of 12.2 months for the modified FOLFIRINOX regimen among older adults ≥75 years with mPC [29]; Berger et al. reported a median OS of 10.2 months for the FOLFIRINOX regimen with no significant differences between younger and older adults (< or ≥65 years) [30]; and Costa et al. reported median OS ranging from 6.7 months (gemcitabine alone) to 13.8 months (FOLFIRINOX) among older adults ≥65 years with mPC [31]. Finally, we confirmed the lesser efficacy of the gemcitabine alone regimen among older adults with mPC. However, our study results should be taken with caution since, due to the retrospective design of our study, the dose-intensity related feasibility of each regimen was not known [32].

The main limitation of our study was the retrospective design with a single centre, which could have led to a patient selection bias, but the consecutive inclusion aimed to reduce this bias. In addition, we considered several clinical and biological parameters to reduce confounding biases. However, further prospective studies are needed to validate our study results using a standardized geriatric assessment [33].

On the basis of our study results, we support the use of chemotherapy, mainly the GnP regimen, among older adults with mPC. Conversely, patients with dependency or walking limitations would probably benefit more from exclusively supportive care, including nutritional and functional rehabilitation. This strategy could be put into perspective with the simple GRADE score (based on weight loss, gait speed, cancer site and cancer extension) which we recently published to help in cancer-treatment decisions concerning older adults and to limit situations of over- or under-treatment [34].

## 5. Conclusions

While the GnP regimen and anti-diabetic drugs were significant protective factors for death, poor functional status and the number of metastatic sites were significant risk factors for death among older adults with metastatic pancreatic cancer.

## Figures and Tables

**Figure 1 cancers-14-01105-f001:**
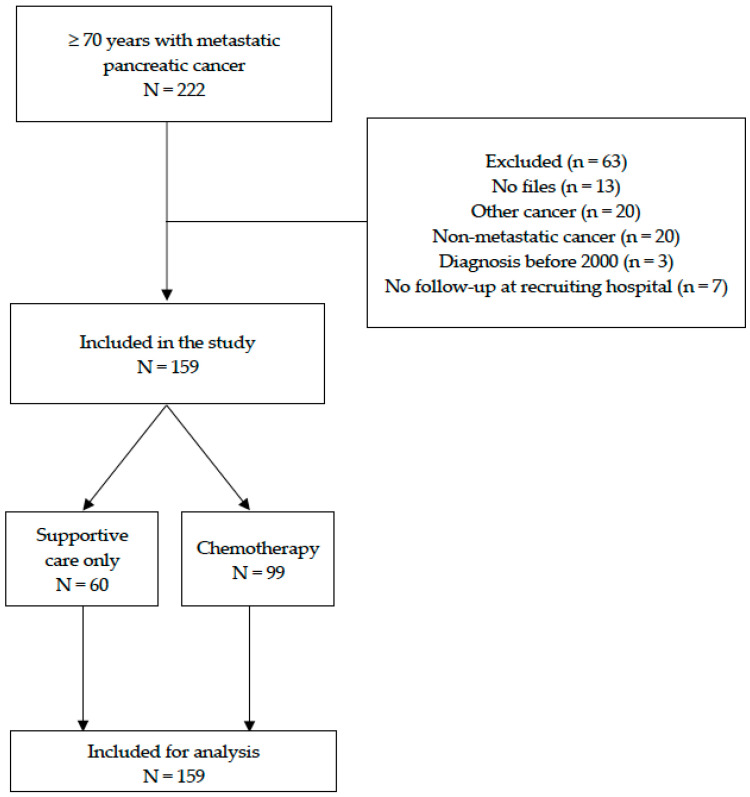
Flow chart for the selection process.

**Figure 2 cancers-14-01105-f002:**
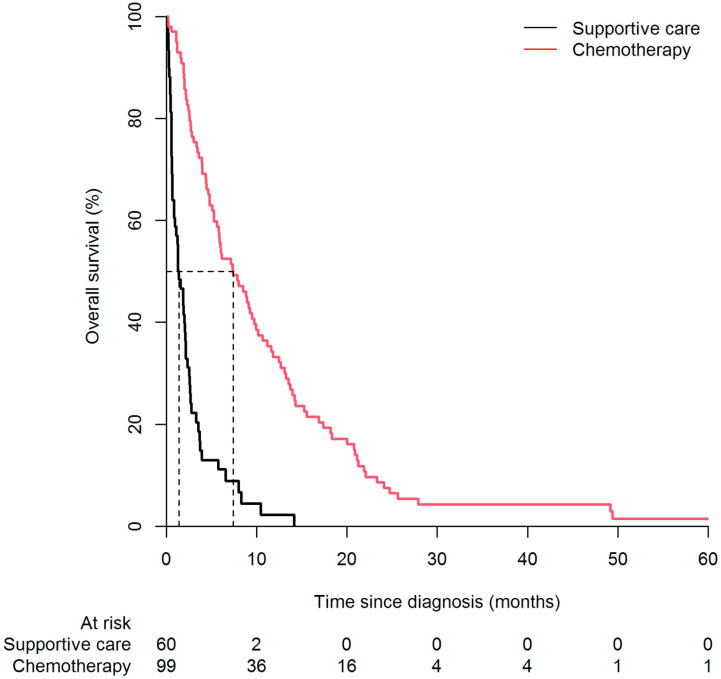
Kaplan-Meier survival curve according to chemotherapy status among older adults with mPC. Dotted lines = median survival in each group.

**Figure 3 cancers-14-01105-f003:**
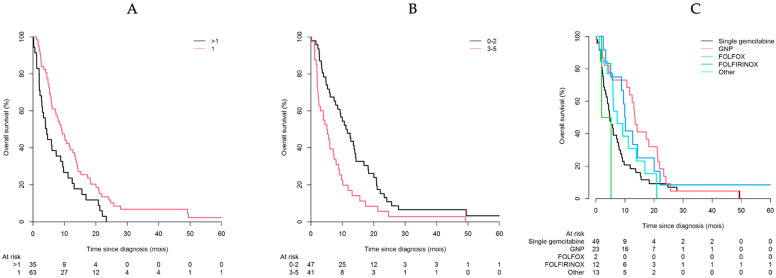
Kaplan-Meier survival curve according to the number of metastatic sites (**A**), ECOG-PS (**B**), and chemotherapy regimens (**C**) among older adults with mPC.

**Table 1 cancers-14-01105-t001:** Baseline characteristics and comparison between patients with chemotherapy and exclusively supportive care among 159 older adults with metastatic pancreatic cancer.

Variables	Whole Cohort	Chemotherapy	Supportive Care	*p* *
	*n* = 159 (%)	*n* = 99 (%)	*n* = 60 (%)	
Age (y), median (min-max)	80.0 (70.0–98.0)	77.0 (70.0–93.0)	83.0 (70.0–98.0)	**<0.001**
Gender (female)	83 (52)	55 (56)	28 (47)	0.27
Cancer site (*n* = 158)				
Head	76 (48)	53 (53.5)	23 (38)	0.06
Body	37 (23)	25 (25)	12 (20)	0.44
Tail	41 (26)	22 (22)	19 (32)	0.18
Unspecified	4 (2.5)	1 (1)	3 (5)	0.15
Metastases (*n* = 153)				
N° of metastasis sites, median (min-max)	1.0 (1.0–4.0)	1.0 (1.0–4.0)	1.0 (1.0–3.0)	0.9
Lymph nodes	20 (13)	12 (12)	8 (13)	0.8
Liver	115 (72)	66 (67)	49 (82)	**0.04**
Lung	27 (17)	21 (21)	6 (10)	0.06
Peritoneal carcinomatosis	43 (27)	30 (30)	13 (22)	0.2
Bone	9 (6)	5 (5)	4 (7)	0.7
Ascites (yes), *n* = 145	25 (17)	13 (14)	12 (22)	0.22
Endobiliary prosthesis (yes), *n* = 156	40 (26)	28 (29)	12 (20)	0.24
Ca-19.9 (KUI/L), median (min-max), *n* = 116	20.0 (0.0–71.0)	18.0 (0.0–26.5)	77.5 (0.0–71.0)	0.11
Marital status (single), *n* = 130	65 (50)	34 (42)	31 (64)	**0.01**
Alcohol consumption (yes), *n* = 123	12 (10)	8 (11)	4 (8)	0.55
Active smoker (yes), *n* = 123	37 (30)	23 (32)	14 (27)	0.51
ECOG-PS, median (min-max), *n* = 141	3.0 (1.0–5.0)	3.0 (1.0–4.0)	3.0 (1.0–5.0)	**<0.0001**
Comorbidities (*n* = 158)				
Charlson’s index ≥ 1	80 (51)	51 (51.5)	29 (49)	0.77
Diabetes	34 (21.5)	26 (26)	8 (14)	0.06
Heart failure	53 (33.5)	30 (30)	23 (39)	0.26
Kidney failure	2 (1)	0 (0)	2 (3)	0.14
Respiratory failure	2 (1)	0 (0)	2 (3)	0.14
Hepatic failure	0 (0)	-	-	-
Stroke	8 (5)	5 (5)	3 (5)	1
Poly-medication (≥5 drugs a day), *n* = 155	58 (37)	39 (40)	19 (33)	0.42
ADL-dependency (≤5/6), *n* = 136	24 (18)	2 (2)	22 (45)	**<0.0001**
Walking limitations (yes), *n* = 130	21 (16)	5 (6)	16 (33)	**<0.0001**
Malnutrition				
Weight loss ≥ 5%, *n* = 119	79 (66)	48 (61.5)	31 (76)	0.12
BMI < 21 kg/m^2^, *n* = 105	38 (36)	31 (40)	7 (26)	0.19
Albumin < 35 g/L, *n* = 107	71 (66)	43 (59)	28 (82)	**0.01**
CAR, median (min-max), *n* = 82	1.02 (0.00–33.3)	0.80 (0.00–33.3)	2.60 (0.10–17.2)	**0.001**
Neutrophil cell count, median (min-max), *n* = 144	6.20 (1.50–33.3)	5.80 (1.5–29.0)	7.70 (2.50–33.3)	**<0.0001**
NLR, median (min-max), *n* = 144	5.40 (0.30–88.0)	4.40 (0.30–88.0)	7.60 (1.50–46.5)	**<0.0001**
Depression (yes), *n* = 156	22 (14)	15 (15)	7 (12)	0.62
Cognitive impairment (yes), *n* = 155	14 (9)	6 (6)	8 (14)	0.09
Haemoglobin < 12 g/dL, *n* = 145	76 (52)	43 (48)	33 (60)	0.15
Total bilirubin (μmol/L), median (min-max), *n* = 139	16.0 (0.00–545)	14.0 (0.00–545)	26.5 (5.00–466)	**0.004**
Anti-diabetic therapy, *n* = 34				0.37
None	5 (15)	3 (11.5)	2 (25)	
Insulin alone	5 (15)	3 (11.5)	2 (25)	
Oral anti-diabetic drugs alone	19 (56)	15 (58)	4 (50)	
Insulin + oral anti-diabetic drugs	5 (15)	5 (19)	0 (0)	
Chemotherapy regimens	-		-	
Gemcitabine alone		49 (49)		
GnP		23 (23)		
FOLFOX		2 (2)		
FOLFIRINOX		12 (12)		
Other **		13 (13)		

* *p* value for chi2 test or Wilcoxon’s test as appropriate; Bold = significant *p* value at a threshold of 5%; ADL = activities of daily living; BMI = body mass index; CAR = CRP to albumin ratio; ECOG-PS = eastern cooperative oncology group performance status; NLR = neutrophil to lymphocyte ratio; GnP = gemcitabine + nab-paclitaxel; FOLFOX = 5-fluorouracil/leucovorin + oxaliplatin; FOLFIRINOX = fluorouracil/leucovorin + irinotecan + oxaliplatin; ** = 5-FU or capecitabine alone.

**Table 2 cancers-14-01105-t002:** Factors associated with overall survival in the whole cohort of older adults with mPC.

Variables	Univariate Analysis			Multivariate Model 1			Multivariate Model 2		
	HR	[95%CI]	*p* *	HR	[95%CI]	*p* *	HR	[95%CI]	*p* *
Age	1.04	[1.01–1.07]	**0.009**	-	-	-	-	-	-
Gender (female)	0.82	[0.59–1.13]	0.21						
Cancer site									
Head	0.55	[0.40–0.77]	**<0.0001**	-	-	-	-	-	-
Body	1.06	[0.72–1.55]	0.77						
Tail	1.55	[1.07–2.26]	**0.02**	-	-	-	-	-	-
Unspecified	6.39	[2.30–17.7]	**<0.0001**	-	-	-	-	-	-
Metastases									
N° of metastasis sites	1.62	[1.23–2.14]	**0.001**	1.37	[0.90–2.07]	0.13	1.44	[0.96–2.12]	0.07
Lymph nodes	1.05	[0.63–1.74]	0.85						
Liver	1.52	[1.05–2.18]	**0.02**	1.71	[1.06–2.77]	**0.02**	1.61	[1.00–2.55]	**0.04**
Lung	0.81	[0.53–1.23]	0.32						
Peritoneal carcinomatosis	1.12	[0.78–1.61]	0.54						
Bone	1.49	[0.75–2.93]	0.25						
Other	1.6	[0.88–2.89]	0.12	0.4	[0.13–1.16]	0.09	0.39	[0.13–1.15]	0.09
Ascites (yes)	1.21	[0.77–1.91]	0.4						
Endo-biliary prosthesis (yes)	0.73	[0.50–1.06]	0.09	-	-	-	-	-	-
Ca-19.9 (kUI/L)	1	[1.00–1.00]	0.07	-	-	-	-	-	-
Marital status (alone)	1.29	[0.90–1.84]	0.16	-	-	-	-	-	-
Alcohol consumption (yes)	1.34	[0.73–2.44]	0.35						
Active smoker (yes)	0.95	[0.63–1.41]	0.78						
ECOG-PS	1.91	[1.53–2.37]	**<0.0001**	1.47	[1.09–1.99]	**0.01**	1.45	[1.08–1.93]	**0.01**
Comorbidities									
Charlson’s index	1.03	[0.92–1.15]	0.59						
Diabetes	0.56	[0.37–0.85]	**0.006**	0.43	[0.24–0.77]	**0.004**			
Heart failure	1.03	[0.73–1.45]	0.86						
Kidney failure	3.98	[0.97–16.3]	0.05	-	-	-	-	-	-
Respiratory failure	2.19	[0.54–8.90]	0.27						
Stroke	0.82	[0.38–1.77]	0.62						
N° of medications	0.99	[0.94–1.05]	0.76						
ADL-scale	0.65	[0.56–0.74]	**<0.0001**	-	-	-	-	-	-
Walking limitations (yes)	3.57	[2.08–6.13]	**<0.0001**	-	-	-	-	-	-
Malnutrition									
Weight loss ≥ 5%	1.24	[0.84–1.84]	0.27						
BMI < 21 kg/m^2^	0.88	[0.59–1.33]	0.57						
CAR ≥ 1.02	1.48	[0.94–2.33]	0.09	-	-	-	-	-	-
Neutrophil (G/L)	1.1	[1.06–1.14]	**<0.0001**	1.12	[1.05–1.20]	**0.001**	1.12	[1.04–1.18]	**0.001**
NLR, median (min-max)	1.02	[1.00–1.03]	**0.02**	0.98	[0.96–1.01]	0.16	0.98	[0.95–1.01]	0.16
Depression (yes)	1.19	[0.74–1.92]	0.47						
Cognitive impairment (yes)	1.56	[0.86–2.84]	0.14	-	-	-			
Haemoglobin level (g/dL)	0.97	[0.87–1.08]	0.52						
Total bilirubin (μmol/L)	1	[1.00–1.00]	0.6						
Anti-diabetic therapy (yes)	0.56	[0.36–0.87]	**0.01**				0.43	[0.23–0.81]	**0.009**
Chemotherapy (yes)	0.23	[0.16–0.33]	<0.0001	0.22	[0.12–0.41]	**<0.0001**	0.23	[0.12–0.41]	**<0.0001**

* *p* value for log-rank test; Bold = significant *p* value at the threshold of 5%; ADL = activities of daily living; BMI = body mass index; CAR = CRP to albumin ratio; ECOG-PS = eastern cooperative oncology group performance status; NLR = neutrophil to lymphocyte ratio.

**Table 3 cancers-14-01105-t003:** Factors associated with overall survival in the chemotherapy subgroup of older adults with mPC.

Variables	Univariate Analysis			Multivariate Model 1			Multivariate Model 2		
	HR	[95%CI]	*p* *	HR	[95%CI]	*p* *	HR	[95%CI]	*p* *
Age	1.01	[0.97–1.04]	0.79						
Gender (female)	0.86	[0.57–1.30]	0.48						
Cancer site									
Head	0.59	[0.39–0.89]	**0.01**	-	-	-	-	-	-
Body	1.19	[0.75–1.91]	0.45						
Tail	1.66	[1.02–2.73]	**0.04**	1.69	[0.92–3.11]	0.09	1.67	[0.91–3.03]	0.09
Metastases									
N° of metastasis sites	2.04	[1.43–2.91]	**<0.001**	1.86	[1.17–2.95]	**0.008**	1.89	[1.43–2.34]	**0.007**
Lymph nodes	0.97	[0.50–1.88]	0.93						
Liver	1.21	[0.78–1.86]	0.39						
Lung	0.91	[0.56–1.49]	0.72						
Peritoneal carcinomatosis	1.43	[0.91–2.24]	0.11	-	-	-	-	-	-
Bone	2	[0.81–4.96]	0.13	-	-	-	-	-	-
Other	2.4	[1.15–5.01]	**0.01**	-	-	-	-	-	-
Ascites (yes)	1.02	[0.55–1.89]	0.95						
Endo-biliary prosthesis (yes)	0.89	[0.56–1.39]	0.6						
Ca-19.9 (kUI/L)	1	[1.00–1.00]	0.37						
Marital status (single)	0.98	[0.62–1.56]	0.94						
Alcohol consumption (yes)	1.75	[0.82–3.73]	0.15	-	-	-	-	-	-
Active smoker (yes)	1.05	[0.63–1.77]	0.84						
ECOG-PS	1.69	[1.22–2.33]	**0.002**	1.74	[1.20–2.53]	**0.003**	1.75	[1.38–2.12]	**0.003**
Comorbidities									
Charlson’s index	0.98	[0.83–1.15]	0.77						
Diabetes	0.6	[0.37–0.99]	**0.04**	0.57	[0.29–1.16]	0.12			
Heart failure	0.83	[0.53–1.30]	0.41						
Stroke	0.69	[0.25–1.89]	0.47						
N° of medications	0.98	[0.92–1.06]	0.65						
ADL-scale	0.4	[0.19–0.83]	**0.01**	-	-	-	-	-	-
Walking limitations (yes)	6.07	[2.02–18.2]	**0.001**	-	-	-	-	-	-
Malnutrition									
Weight loss ≥ 5%	1.05	[0.66–1.68]	0.83						
BMI < 21 kg/m^2^	1	[0.62–1.59]	0.98						
CAR ≥ 1.02	1.41	[0.78–2.54]	0.25						
Neutrophil (G/L)	1.07	[1.00–1.15]	**0.05**	1.06	[0.99–1.13]	0.12	1.05	[0.99–1.10]	0.1
NLR, median (min-max)	1.02	[1.00–1.05]	0.1	-	-	-	-	-	-
Depression (yes)	1.23	[0.68–2.22]	0.5						
Cognitive impairment (yes)	1.17	[0.47–2.90]	0.73						
Haemoglobin level (g/dL)	1.01	[0.87–1.17]	0.89						
Total bilirubin (μmol/L)	1	[1.00–1.00]	0.95						
Anti-diabetic therapy (yes)							0.49	[0.23–1.03]	0.06
Chemotherapy regimens									
Single gemcitabine	1.00 (ref)	–	-	1.00 (ref)	-	-	1.00 (ref)	-	-
GnP	0.54	[0.32–0.91]	**0.02**	0.47	[0.25–0.89]	**0.02**	0.45	[0.25–0.87]	**0.01**
FOLFOX	2.53	[0.60–10.6]	0.2	3.45	[0.77–15.4]	0.1	3.41	[0.77–15.0]	0.1
FOLFIRINOX	0.55	[0.28–1.08]	0.08	0.97	[0.44–2.15]	0.94	1.01	[0.46–2.20]	0.96
Other **	0.87	[0.46–1.62]	0.65	0.86	[0.39–1.91]	0.71	0.86	[0.39–1.89]	0.71

* *p* value for log-rank test; Bold = significant *p* value at the threshold of 5%; ADL = activities of daily living; BMI = body mass index; CAR = CRP to albumin ratio; ECOG-PS = eastern cooperative oncology group performance status; NLR = neutrophil to lymphocyte ratio; GnP = gemcitabine + nab-paclitaxel; FOLFOX = 5-fluorouracil/leucovorin + oxaliplatin; FOLFIRINOX = fluorouracil/leucovorin + irinotecan + oxaliplatin; ** 5-FU simplified or capecitabine alone.

## Data Availability

The data presented in this study are available on request from the corresponding author.

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
