# Peer review of "Overall Survival and Prognostic Factors among Older Patients with Metastatic Pancreatic Cancer: A Retrospective Analysis Using a Hospital Database"

_cancers, 2022, doi:10.3390/cancers14051105_

Round 1
Reviewer 1 Report
This retrospective study aimed to explore the prognostic factors in elderly patients diagnosed with metastatic pancreatic cancer. The authors examined the relevant clinicopathological parameters in a total of 159 consecutive patients, and finally showed that GnP regimen and anti-diabetic drugs were significantly protective factors, and poor functional status and number of metastatic sites were significant risk factors for death. This study topic, especially as geriatric assessment, is clinically important, and the manuscript is easy to follow. However, I listed some concerns to be discussed as below.
- When examined in regard of chemotherapy or supportive care, we can not deny the possibility of presence for bias between the groups. That is, most patients in supportive care group do not receive chemotherapy, due to worse PS and ADL, or worse nutritional status. Therefore, the comparison such as Figure 2 should be inappropriate.
- Two multivariate analysis models were conducted in Table 2. It is interesting to focus on the diabetes or anti-diabetic therapy as prognostic factors. However, there is a question whether these two factors could be treated similarly. The details for these factors should be added.
- No of metastasis sites was identified as a prognostic factor, but the definition of this factor is unclear. Please clarify it.
- GnP was also identified as a prognostic factor, but other regimens were not. Please discuss and add the comment in the discission section.
Author Response
please find the response point by point
thank you

Reviewer 2 Report
In this article, the authors retrospectively investigated the overall survival of the metastatic pancreatic cancer patients over 70years receiving chemotherapy or best supportive care. They demonstrated that chemotherapy and diabetes (anti-diabetic therapy) was the independent better prognostic factors for the OS in those elder patients and also discovered several independent worse prognostic factors by multivariate analysis. In the chemotherapy receiving cohort, they found that GnP regimen showed better OS compared to the gemcitabine alone and also detected ECOG-PS and number of metastatic sites were independent prognostic factors. As the authors described in the Discussion, similar studies have already exist, but this study focused on the elder patients and also investigated predictive role of geriatric parameters for the OS. I raised a couple of points to be addressed. 1) In the analysis of chemotherapy cohort, the authors had better explain how they analyzed the cohort. Especially, chemotherapy regimens seemed to be compared to gemcitabine monotherapy, but there is no explanation. 2) In the analysis of chemotherapy cohort, three independent prognostic factors were detected. The authors had better present Kaplan-Meier curve for each prognostic factor, respectively, like Figure 2. 3) In the whole cohort study, the authors detected anti-diabetic therapy as an independent prognostic factor for OS. However, they did not include anti-diabetic therapy in the analysis of chemotherapy cohort. Since they discussed it as the main findings of this study, they should include it in the chemotherapy cohort analysis. In addition, they discussed the association of metformin and OS, but they did not check which drugs the patients were taking. If the authors know that majority of the patients were taking metformin, they can describe this study is consistent with previous reports. 4) This study demonstrates that chemotherapy might improve the OS of pancreatic cancer patients over 70 years. Especially for the elder patients, it is important how much dose was administered. Elder patients frequently experienced the dose reduction due to the adverse effects and the reduced dose frequently enabled the patients to continue the chemotherapy. If there is no information of the dose intensity, it can be somewhat misleading.Author Response
please find the reply point by point
thank you

Round 2
Reviewer 1 Report
I have read the revised manuscript throughout and think that it is acceptable for the publication.
Author Response
Thank you very much for your reviewing
Reviewer 2 Report
The authors improved the manuscript according to the points raised by the reviewers. Since this is a single center, middle-sized study, it might not be impossible for authors to check the dose-intensity and also the detail of anti-diabetic drugs. As for metformin, there might be such data that metformin has been the most frequently prescribed for all the diabetic patients in their hospital, etc. The authors might be able to discuss using such data, not by just an assumption from the reference.Author Response
Dear reviewer 2
The authors improved the manuscript according to the points raised by the reviewers. Since this is a single center, middle-sized study, it might not be impossible for authors to check the dose-intensity and also the detail of anti-diabetic drugs. As for metformin, there might be such data that metformin has been the most frequently prescribed for all the diabetic patients in their hospital, etc. The authors might be able to discuss using such data, not by just an assumption from the reference.
We thank the Reviewer 2 for his/her comments
In agreement with the Reviewer 2 we have now retrieved some data regarding anti-diabetic therapy and chemotherapy.
Regarding anti-diabetic therapy, of the 19 diabetic patients which were treated with oral anti-diabetic drugs, data were available for 17 of them. Among these, 14/17 (82%) were treated with metformin, 7/17 (41%) with glinides, and 4/17 (23.5%) with sulfonamides.
We have now added these data as follows:
- Lines 164-167 “Among anti-diabetic treatments, oral anti-diabetic drugs concerned 19/34 (56%) of the diabetic patients. The distribution of oral anti-diabetic drugs was available for 17 of them as follows: 14/17 (82%) were treated with metformin, 7/17 (41%) with glinides, and 4/17 (23.5%) with sulfonamides.
- Lines 246-248 “Indeed, in our study, 82% of the diabetic patients were treated with metformin which is in line with the literature [17,18].”
Regarding chemotherapy, the dose-intensity remained unavailable. Nevertheless, in our center, the oncologists reported no dose reduction for gemcitabine, a 20% dose reduction of nab-paclitaxel for the GnP regimen in 90% of cases, and no loading dose for the remaining regimens (FOLFOX and FOLFIRINOX).
